# Dislodgement resistance and structural changes of tricalcium silicate-based cements after exposure to different chelating agents

Özgür İlke Ulusoy[1☯], Nidambur Vasudev Ballal[2☯], Rajkumar Narkedamalli[2], Nuran Ulusoy[3], Krishna Prasad Shetty[4,5]*, Alexander Maniangat Luke[4,5]

1 Faculty of Dentistry, Department of Endodontics, Gazi University, Ankara, Turkey, 2 Department of Conservative Dentistry and Endodontics, Manipal College of Dental Sciences, Manipal, Manipal Academy of Higher Education, Karnataka, India, 3 Faculty of Dentistry, Department of Restorative Dentistry, Near East University, Nicosia, Northern Cyprus, 4 Department of Clinical Science, College of Dentistry, Ajman University, Ajman, United Arab Emirates, 5 Centre for Medical and Bio-Allied Health Sciences Research, Ajman University, Al-Jruf, Ajman, United Arab Emirates

☯ These authors contributed equally to this work.
* kprasad11@gmail.com

**Data Availability Statement:** All the data will be available after acceptance of the manuscript.

## Abstract

This study aimed to evaluate the dislodgement resistance and structural changes of different mineral trioxide aggregate cements (MTA) like Pro-Root MTA, Ortho MTA, and Retro MTA after exposure to sodium hypochlorite (NaOCl), NaOCl-Ethylenediaminetetraacetic acid (EDTA), 1-hydroxyethylidene-1, 1-bisphosphonate (Dual Rinse HEDP), and NaOCl-Maleic acid (MA). The root canal spaces of 150 dentine slices were obturated using tricalcium silicate cements and divided into 3 groups (n = 50): Group1: ProRoot MTA, Group2: Retro MTA, and Group3: Ortho MTA. The samples in each group were further subdivided into four experimental (n = 10) and one control groups (n = 10): 2.5% NaOCl-17% EDTA, Dual Rinse HEDP, 2.5% NaOCl-7% Maleic acid, 2.5% NaOCl, distilled water (control). The dislodgement resistance and structural changes of cements were measured. Use of DR HEDP resulted in higher dislodgement resistance compared to 17% EDTA and 7% MA in the samples obturated with Ortho MTA and Pro-Root MTA (p<0.001). In Retro MTA group, samples treated with DR HEDP and 17% EDTA had higher dislodgment resistance compared to 7% MA (p<0.001). On microstructural and elemental analysis of all the three MTA cements, samples treated with 17% EDTA and 7% MA were more amorphous and granular when compared to DR HEDP, which was pettle shaped. Calcium level was decreased more in samples treated with 17% EDTA and 7% MA when compared to DR HEDP.

## Introduction

Tricalcium silicate-based cements are widely used in various endodontic procedures including apexification, vital pulp therapy, root canal perforation repairs, and periapical surgery due to their biocompatibility, hard-tissue stimulation ability, and enhanced sealing property [1,2].

**Funding:** The authors received no specific funding for this work.

**Competing interests:** The authors have declared that no competing interests exist.

ProRoot MTA (Dentsply, Tulsa Dental Products, OK, USA), the first introduced tricalcium silicate-based cement, consists of Tricalcium silicate, Dicalcium silicate, Tricalcium aluminate, Tetracalcium aluminoferrite, Gypsum, Free calcium oxide, and Bismuth oxide. It has been suggested to have some drawbacks such as long setting time, tooth discoloration, difficult manipulation, and high cost [3]. Ortho-MTA (BioMTA, Seoul, Korea), and Retro-MTA (BioMTA) have been introduced to overcome these shortcomings of ProRoot MTA. Ortho-MTA and ProRoot MTA, have gained high popularity in endodontic therapy due to their advanced properties [4]. Retro-MTA is a hydraulic bioceramic material, in which Portland cement was removed from its composition and replaced by a new generation nanomaterial, which resulted in elimination of toxic compounds from the composition of MTA [5]. It consists of calcium carbonate, silicon dioxide, aluminum oxide and zirconium oxide. It was reported that Retro-MTA has a shorter setting time compared to the ProRoot MTA and has no discoloration potential, due to the presence of zirconium oxide as an alternative radiopacifier to bismuth oxide [6]. Ballal et al. reported that Retro-MTA healed pulp tissue faster than ProRoot MTA when used as a direct pulp capping agent [7]. Ortho-MTA has a similar composition to ProRoot MTA. However, it has been reported to have less heavy metal content compared to ProRoot MTA and has shown to have similar dislodgement resistance with Pro Root MTA [8,9]. It has also been demonstrated that ProRoot MTA contained traces of arsenic (1.16 ppm), whereas Ortho-MTA did not [8].

Application of 5.25% NaOCl followed by 17% EDTA is the most widely used irrigation regimen aiming at smear layer removal, antimicrobial disinfection, and tissue dissolution. However, in the combined use of NaOCl and EDTA, antibacterial and tissue dis-solution capacities of NaOCl can be diminished due to the reduced free chlorine [10]. In addition, EDTA could not efficiently remove the smear layer from the apical thirds of root canals [11]. Mixture of NaOCl and HEDP (etidronic acid) has recently been suggested to overcome this undesired interaction between EDTA and NaOCl, and to achieve effective organic tissue dissolution and smear layer removal in the root canals [12–14]. As of 2016, a commercial CE-marked HEDP product for endodontic usage, Dual Rinse HEDP (Medcem GmbH, Weinfelden, Switzerland), has become available [15]. This is supposed to be added to the NaOCl solution for clinical use, immediately before treatment to receive an all-in-one root canal irrigant with combined proteolytic and chelating properties. This Dual Rinse HEDP was found to be biocompatible [16] and has antimicrobial activity similar to that of NaOCl [17]. Maleic acid (MA) is a mild organic acid and was suggested to be another potential alternative to EDTA [18]. It was reported that 7% MA had enhanced smear layer removal capacity compared to 17% EDTA at the apical third of root canal system [11] and was less cytotoxic [19].

Previous studies have demonstrated the negative impact of conditioning the root canal walls with these chelating agents, on the structural changes and adhesive capacity of tri calcium silicate-based cements to root dentine [20–23].

However, following repair of root canal perforations, or placement of apical plugs using different types of MTA cements, the rest of the root canal cavity is generally obturated using gutta-percha cones and resin or bioceramic based sealers, once the MTA cement is completely set. Prior to the root canal obturation, canals should be finally irrigated using NaOCl and different chelating agents to eliminate smear layer. During this process, chelating agents will come in contact with the set MTA cement used for apical plugs or perforation repair. To date, no comparative studies are available that evaluate the effect of novel chelating agents on the properties of completely set MTA cements. Hence, the aim of this study was to evaluate the dislodgement resistance of ProRoot MTA, Ortho-MTA, and Retro-MTA from the root canal dentine and its structural changes after exposure to 2.5% NaOCl, 2.5% NaOCl-17% EDTA, Dual Rinse HEDP, and 2.5% NaOCl-7% MA. The null hypothesis tested was that there is no

difference in the potential of 17% EDTA, 7% MA, 2.5% NaOCl and Dual Rinse HEDP in reducing the dislodgment resistance and changing the microstructure of different MTA cements used.

## Materials & methods

### Sample size estimation

The number of samples included in the present study was determined based on previous research [24]. Based on this research, sample size was estimated at 95% confidence interval and with the power of 80% which resulted in 10 samples in each group. However, in the microstructural analysis experiment, since the data was of qualitative nature than quantitative, the experiments were performed in triplicates (n = 3).

### Specimen preparation

**Evaluation of dislodgment resistance.** Ethical clearance was (335/2019) obtained from the institutional review board (Kasturba Medical College-Kasturba Hospital, Manipal, India) for the use of human extracted teeth for this study. Since this study used anonymized human extracted teeth, no consent from patient was required. A total of 150 single-rooted human teeth were selected. Teeth with caries, root cracks, immature apices and previous endodontic treatment were excluded. Soft tissue fragments and calcified debris on the specimens were removed using ultrasonic scalers. The specimens were stored in a solution of 0.2% sodium azide (Sigma-Aldrich, Steinheim, Germany) at 4°C until use [25]. Radiographs of the specimens were taken from buccal and mesial aspect to confirm a straight, single canal with mature apices and without any calcifications. The teeth were decoronated using a diamond disc (Horico, Berlin, Germany) under water spray. Each root was embedded in cold cure acrylic resin (Dentsply Sirona Endodontics, Ballaigues, Switzerland) and sectioned horizontally in the middle third using hard tissue microtome (Leica Biosystems, Nussloch GmbH, Germany) under continuous water cooling to obtain a slice of 2.0±0.1 mm thickness. The thickness of each slice was confirmed using a digital caliper (Sigma-Aldrich, Darmstadt, Germany). In each slice, the root canal space was enlarged to a size 4 peeso reamer (Dentsply Sirona). The root canal diameter for each slice was also measured using digital caliper and recorded. All the root slices were placed in orbital shaker (ThermoFisher, Scientific, Waltham, Massachusetts, USA) and treated with 2.5% NaOCl (Vista Dental Inc., Racine, WI, USA) for 5 minutes to remove any debris and randomly divided into 3 main groups (n = 50) according to the tricalcium silicate filling material used: Group 1: ProRoot MTA (Dentsply), Group 2: Retro-MTA (BioMTA), and Group 3: Ortho-MTA (BioMTA). The three tricalcium silicate cements were mixed according to manufacturers' instructions and then filled into the dentin slices. The samples were wrapped in wet gauze and allowed to set in 100% humidity for 48 hours. The filled samples in each group were further subdivided into four experimental (n = 10) and one control group (n = 10), immersed a beaker containing one of the test solutions placed in orbital shaker: (1) 2.5% NaOCl-17% EDTA (Vista Dental) (2) Dual Rinse HEDP (Medcem), (3) 2.5% NaOCl-7% MA (Sigma Aldrich, St. Louis, MO, USA), (4) 2.5% NaOCl (Vista Dental), and (5) distilled water (control). In group 2, one capsule of Dual Rinse HEDP (containing 0.9 g of etidronate powder), was directly mixed with 10 mL of 2.5% NaOCl solution (Vista Dental) before treatment. The exposure time of each specimen to the experimental solutions was 1 minute. After 1 minute of treatment, all the specimens were rinsed with distilled water and air-dried.

**Push-out bond strength measurement.** Push-out bond strength test was carried out using a universal testing machine (Instron, Massachusetts, USA). The force at a crosshead speed of 1 mm/min was applied from the apical to the coronal direction using stainless steel

plungers with 0.6 mm diameter, which was positioned to contact the entire surface of filling material. The maximum force (F) applied when the bond failure occurred was recorded in Newton.

The push-out bond strength was calculated in MPa using the formula given below:

Push out bond strength (MPa) = Force (N)/adhesion surface area (mm$^2$)

The adhesion surface area was calculated by the following equation:

Adhesion surface area (mm$^2$) = 2 × π × r × h, where π is the constant 3.14, r is radius of the root canal preparation and h is the thickness of the root slice.

**Fractographic analysis.**  All samples from each group were subjected to stereomicroscopic analysis (Olympus SZX 61, Japan) at 40X magnification for the evaluation of bond failures (Fig 1). The modes of bond failures were categorized as:

a.  adhesive failure; between root canal dentine wall and cement interface

b.  cohesive failure; within the cement

c.  mixed failure; both cohesive and adhesive

## Microstructure analysis

**Scanning electron microscopy and energy dispersive spectroscopy.**  Fifteen circular shaped specimens (2 mm thickness, 8 mm diameter) of ProRoot MTA (Dentsply), Retro-MTA (BioMTA) and Ortho-MTA (BioMTA) were prepared using a putty mold. The specimens were randomly divided into five groups (n = 3) and treated with the test irrigants in the same manner described for dislodgment resistance testing. The specimens were then rinsed with distilled water and dehydrated subsequently using ascending grades of ethanol (25%, 50%, 75% and 100%) for 15 min each, mounted on metal stubs, coated with gold using an ion-sputtering machine and examined with a scanning electron microscope and energy dispersive spectrometer (SEM-EDX; JSM-6010, JEOL). Images were taken to identify the surface characteristics and elemental analysis of different MTA specimens at 3000X magnification and 10 kV.

All statistical analysis was performed using SPSS (The statistical package for social sciences) Version 15.0 (SPSS Inc, Chicago, IL, USA). The normality of the data was tested using Kolmogorov-Smirnov Test. Since the data was not normally distributed, it was presented as median and interquartile ranges. Kruskal-Wallis test was used for group comparisons. And Mann-Whitney U-test was used for the intergroup comparisons. Type of bond failures were assessed by Chi-square test. P<0.05 was considered to be significant.

## Results

### Dislodgment resistance

The dislodgement resistance values of different MTA cements after irrigation with different irrigation protocols are summarized in Table 1. The modes of failures after dislodgement of filling materials are shown in Table 2. The Kruskal-Wallis test showed that there were significant differences between the groups (p<0.0001). Mann-Whitney U-test revealed that use of DR HEDP resulted in higher dislodgement resistance values compared to the 17% EDTA and 7% MA in the samples obturated with Ortho-MTA and Pro-Root MTA (p<0.001). There was no significant difference between 17% EDTA (p = 0.45) and 7% MA (p = 0.05). In Retro-MTA group, samples treated with DR HEDP and 17% EDTA had higher dislodgment resistance compared to 7% MA (p<0.001). There was no significant difference between DR HEDP and 17% EDTA (p = 0.14). Regarding the effect of 2.5% NaOCl in Ortho-MTA group, samples

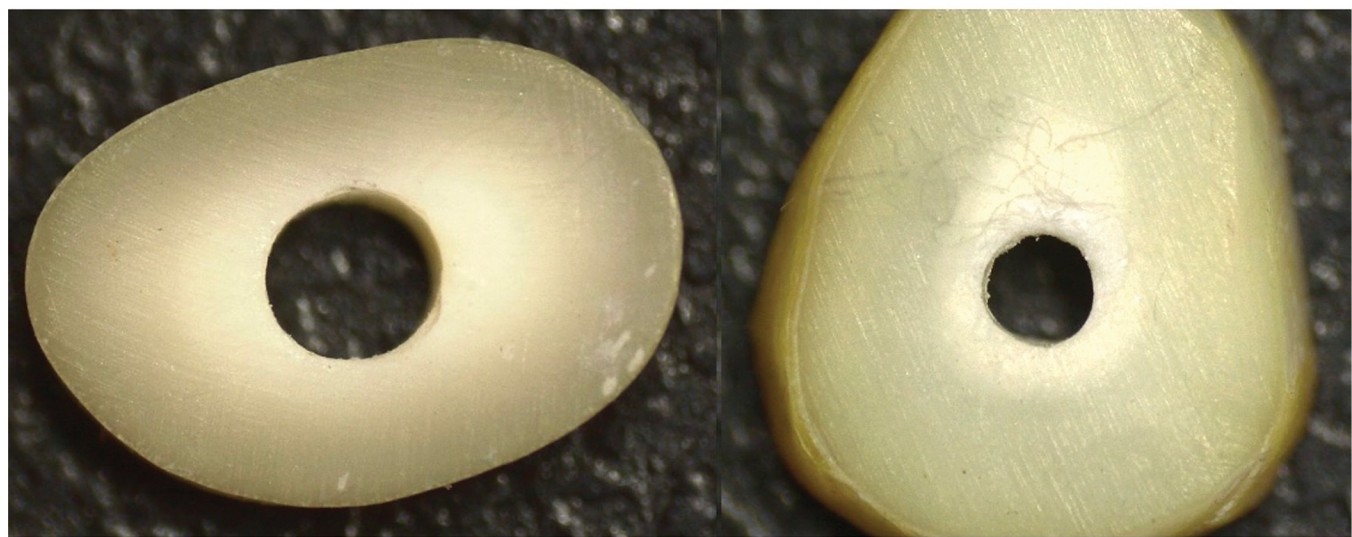

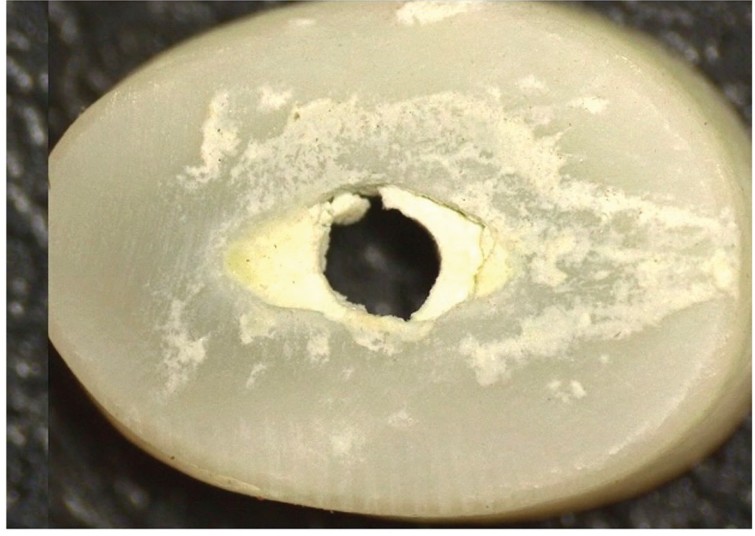

**Fig 1. Representative stereomicroscopic images of the types of bond failures.** (A) Adhesive failure, (B) Cohesive failure, and (C) Mixed failure.

reated with 2.5% NaOCl had similar dislodgment resistance as 17% EDTA (p = 0.27) and 7% MA (p = 0.05), which was inferior to DR HEDP (p = 0.02). In Pro-Root MTA group, 2.5% NaOCl had similar effect as that of DR HEDP (p = 0.01) which was superior to 17% EDTA (p = 0.00) and 7% MA (p = 0.13). In Retro-MTA group, dislodgment resistance of samples treated with 2.5% NaOCl was equivalent to 17% EDTA (p = 0.49) and DR HEDP (p = 0.94) which was superior to 7% MA (p = .04). Use of distilled water resulted in the highest dislodgement resistance of all the three different types of MTA cements used (p<0.001). Irrespective of the chelators, the highest resistance to dislodgement was derived from the samples obturated using Pro-Root MTA (p<0.001), although the difference was not

**Table 1. Dislodgement resistance.**

| Filling material | Ortho-MTA | | ProRoot MTA | | Retro-MTA | | p-value |
|---|---|---|---|---|---|---|---|
| Irrigant | Median | IQR | Median | IQR | Median | IQR | |
| Maleic acid | 1.750 A, a | 0.78 | 5.345 AB, b | 1.09 | 1.845 A, a | 1.58 | <0.001*** |
| Dual Rinse HEDP | 3.925 B, a | 1.81 | 7.225 C, b | 4.56 | 4.155 B, a | 3.34 | 0.002** |
| EDTA | 2.000 A, a | 1.66 | 4.445 A, b | 2.95 | 3.785 B, b | 2.19 | 0.007** |
| NaOCl | 2.595 A, a | 1.86 | 6.400 CB, b | 3.59 | 4.930 B, ab | 12.51 | 0.008** |
| Distilled water | 6.060 C, a | 2.35 | 11.400 D, b | 5.2 | 12.160 C, b | 2.66 | 0.002** |
| p-value | <0.001 | | <0.001 | | <0.001 | | |

Push-out bond strength (Median and Interquartile range) of Ortho MTA, Pro-Root MTA and Retro MTA cements after treatment with different experimental solutions. Identical uppercase letters show no significant differences between the irrigation solutions, and identical lowercase letters show no significant differences between the MTA cements (Kruskal-Wallis and Mann-Whitney U-tests).

**- highly significant

***- very highly significant.

statistically significant from the Retro-MTA when EDTA was used. There was no significant difference between the failure modes observed in the samples of all the three different types of MTA cements used. Representative pictures of different types of bond failures in each group are demonstrated in Fig 1.

## Microstructure

The surface structure of ProRoot MTA treated with distilled water (control) was more amorphous. Samples treated with 17 EDTA, and 7% MA was less amorphous compared to distilled water. Samples treated with DR HEDP had pettle-shaped crystals, whereas samples treated with 2.5% NaOCl were more granular in appearance. In Ortho-MTA group, samples treated with distilled water, 17% EDTA, 7% MA and 2.5% NaOCl were amorphous. However, samples treated with DR HEDP had pettle-shaped crystals. All samples in Retro-MTA group irrespective of the irrigants, had amorphous structure. Representative SEM image of microstructure of different MTA cements is demonstrated in Fig 2.

**Table 2. Modes of failure.**

| Ortho MTA (n = 10) | Mode of Failure | 7% Maleic Acid | Dual Rinse HEDP | 17% EDTA | NaOCl | Distilled Water | P-Value |
|---|---|---|---|---|---|---|---|
| | Adhesive | 0 | 2 | 0 | 0 | 3 | 0.119 |
| | Cohesive | 0 | 0 | 0 | 1 | 0 | |
| | Mixed | 10 | 8 | 10 | 9 | 7 | |
| ProRoot MTA (n = 10) | Adhesive | 0 | 2 | 0 | 0 | 3 | 0.257 |
| | Cohesive | 0 | 0 | 0 | 1 | 0 | |
| | Mixed | 10 | 8 | 10 | 9 | 7 | |
| Retro MTA (n = 10) | Adhesive | 0 | 2 | 0 | 0 | 3 | 0.093 |
| | Cohesive | 0 | 0 | 0 | 1 | 0 | |
| | Mixed | 10 | 8 | 10 | 9 | 7 | |

Modes of bond failures within Ortho MTA, Pro-Root MTA and Retro MTA samples treated with different experimental solutions (Chi-square test).

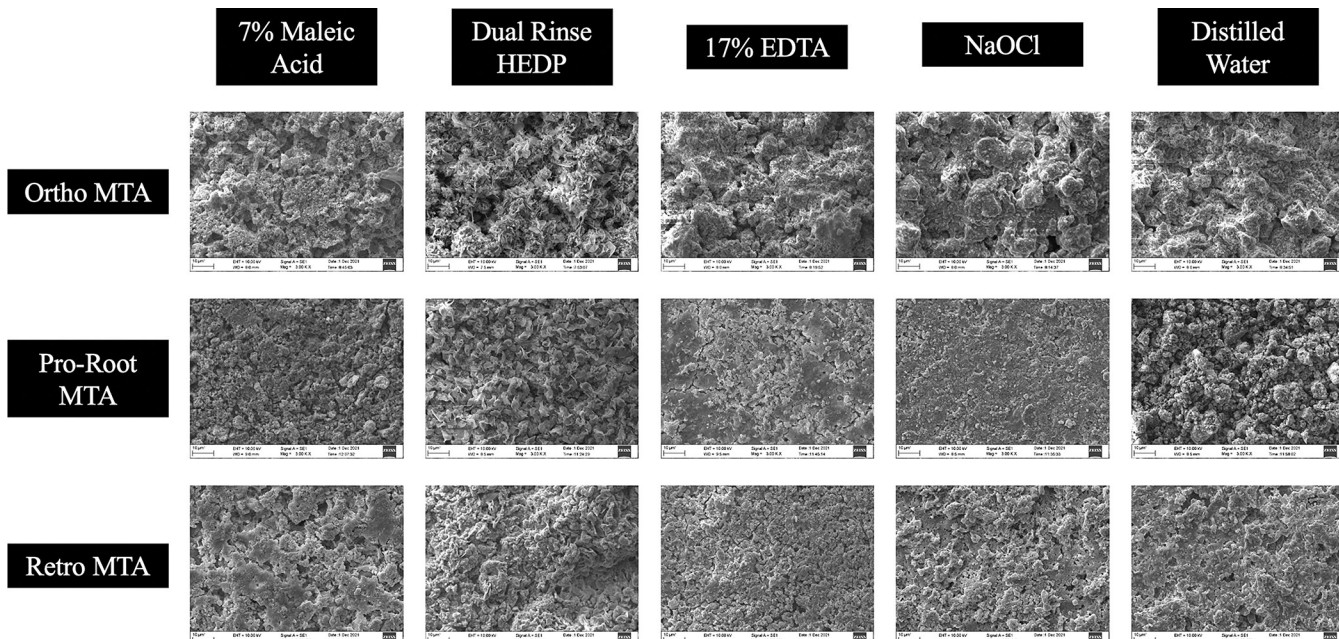

**Fig 2. Scanning electron microscopy images of hydrated, set Ortho-MTA, Pro-Root MTA and Retro-MTA specimens after exposure to 7% MA, DR HEDP, 17% EDTA, 2.5% NaOCl and distilled water (control).**

## Elemental analysis

In all the three types of MTA cements tested, calcium level was decreased more in samples treated with 17% EDTA and 7% MA when compared to DR HEDP. The levels of Carbon, Magnesium, Silica, Oxygen, Aluminium, Sodium, Bismuth and Zirconium was not significantly altered with any of the experimental irrigants used. Representative energy dispersive spectroscopy images of three different types of MTA cements treated with experimental irrigating agents are demonstrated in Figs 3–5.

## Discussion

After placement of apical plugs or repair of the root canal perforations and resorption sites using tricalcium silicate cements, the rest of the root canal system is irrigated with chelating agents for the removal of smear layer before backfilling of the root canals. However, these agents may disrupt the structure of tricalcium silicate cements, thereby reducing the adhesion and dislodgement resistance of them to the root canal dentine. It has previously been suggested that the structure and setting reactions of the tricalcium silicate-based cements can be damaged by the decalcifying agents, which can influence the particle-binding hydration phases and hydration mechanism of MTA [20,21]. This process may result in reduction of the chemical adhesion of MTA to the root canal dentine [26]. It was also reported that demineralizing agents used to remove the smear layer affect the biomineralization process of the tricalcium silicate-based cements by decreasing the calcium ions in the dentine [27].

In the present study, the samples filled with tricalcium silicate-based cements were directly exposed to different chelating agents to simulate the clinical conditions, in which the final irrigation of root canals were performed after repair of perforation site or re-sorption defect. Use of distilled water resulted in the lowest dislodgement resistance of the three different types of filling materials compared to other irrigating solutions used. Hence, the null hypothesis tested

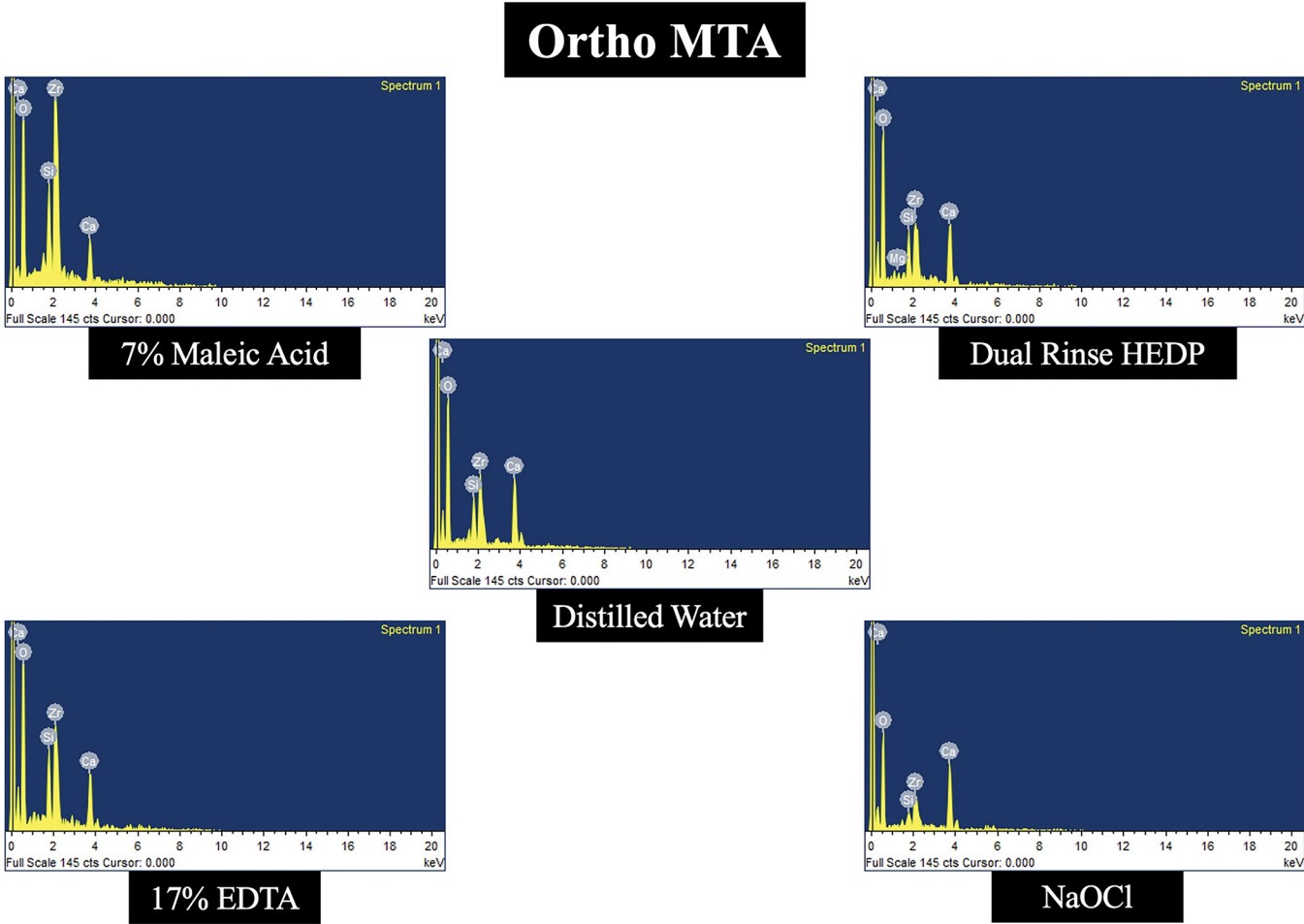

**Fig 3. Representative energy dispersive spectroscopy image of Ortho-MTA cement treated with 7% MA, DR HEDP, 17% EDTA, 2.5% NaOCl and distilled water (control).**

in the present study was rejected. This can be explained by the non-destructive effect of the distilled water on the setting reactions and structure of tricalcium silicate-based cements. However, other irrigation solutions possess some chelating potential that can harm the structure of the MTA cements, thereby decreasing their adhesion to dentine. Additionally, it has previously been reported that the calcium silicate-based cements require moisture in the smear layer for their setting reactions [28,29]. When the tricalcium silicate-based fillings were exposed to the chelating agents, the smear layer that was present beneath the filling could be removed to some extent and this process could have negatively influenced the adhesion of MTA-based cements. The highest push-out strength values obtained from the samples exposed to distilled water can also be explained with this concept.

The results of the present study have also revealed that the dislodgement resistance of different types of tricalcium silicate cements varied after use of different chelating agents. This could be caused by possible interactions between the chelators and filling materials. This result indicates that the type of the chelating agents used for the removal of smear layer from the root canal dentine after perforation repair can be selected considering the type of tricalcium silicate cements to be used. For example, in the present study, irrigation with DR HEDP resulted in higher dislodgement resistance values compared to the EDTA and MA in the samples

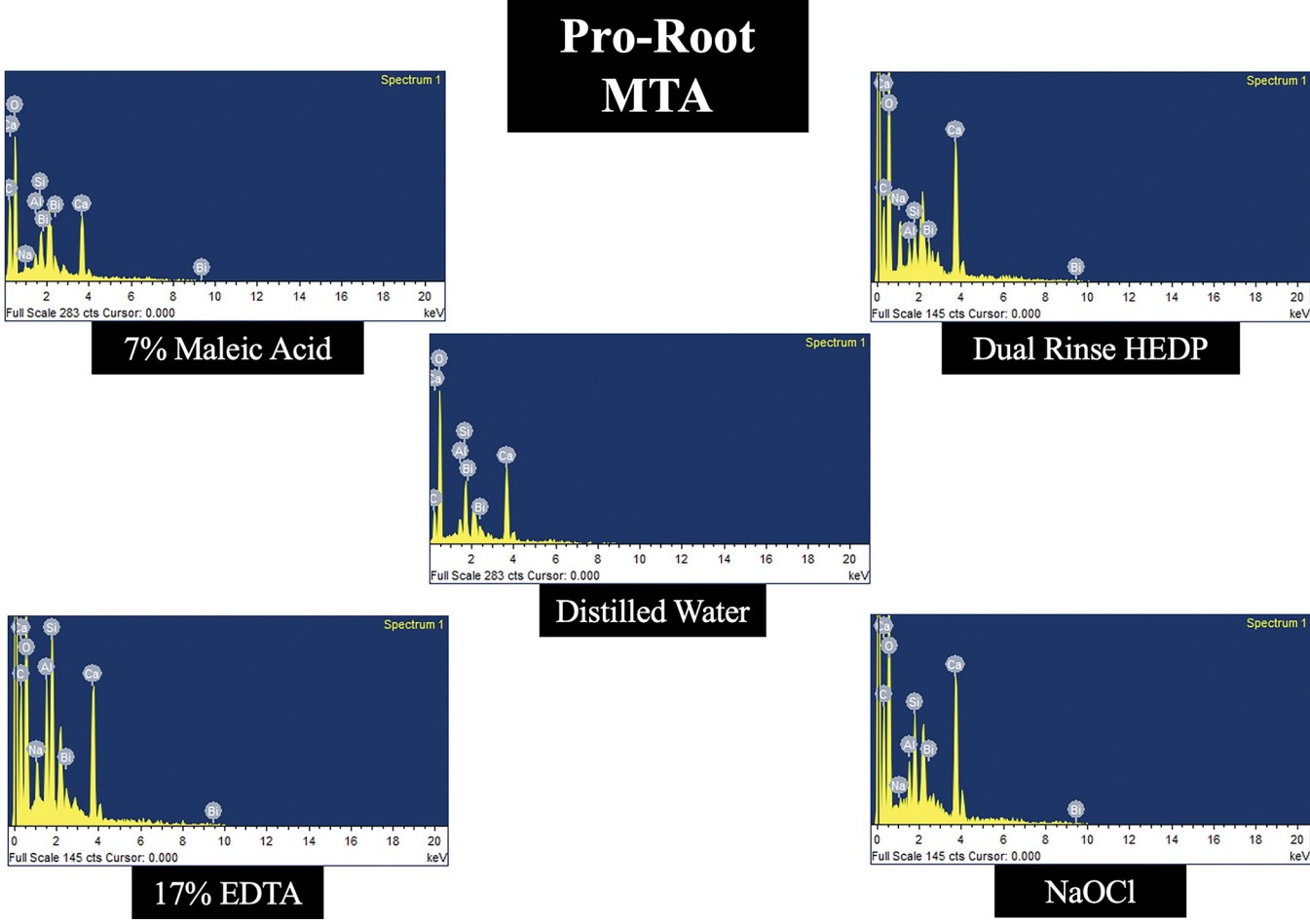

**Fig 4. Representative energy dispersive spectroscopy image of Pro-Root MTA cement treated with 7% MA, DR HEDP, 17% EDTA, 2.5% NaOCl and distilled water (control).**

obturated with Ortho-MTA and Pro-Root MTA. This finding can be attributed to the mild chelating potential of HEDP which does not harm the structure of the tricalcium silicate cements. Unlike Pro-Root MTA and Ortho-MTA, samples in Retro-MTA group treated with EDTA had similar dislodgment resistance as that of DR HEDP. The exact reason for the better dislodgment resistance with EDTA is not known. It might be attributed to the compositional difference of Retro-MTA cement which consists of Zirconia as a radio opacifying agent when compared to Bismuth oxide as seen in Pro-Root and Ortho-MTA cements. Mild chelating property of DR HEDP was also witnessed in the present study, by the surface alterations in the microstructure of ProRoot and Ortho-MTA cements, which had pettle shaped when compared to EDTA and MA treated samples, which had amorphous and porous surface structure. The less crystalline and more porous surface structure of MTA cements, treated with strong chelators like EDTA or MA is probably due to the dissolution of the ettringite and portlandite interlocking crystalline components. These crystalline phases are important in interlocking the entire mass of the set, hydrated material [28]. Also, in the elemental analysis of the MTA specimens, it was observed that the calcium level was reduced in all the three types of MTA cements treated with EDTA and MA. This again reinforces the fact that strong chelators like EDTA and MA are detrimental to the microstructure of MTA cements. On the other hand, DR

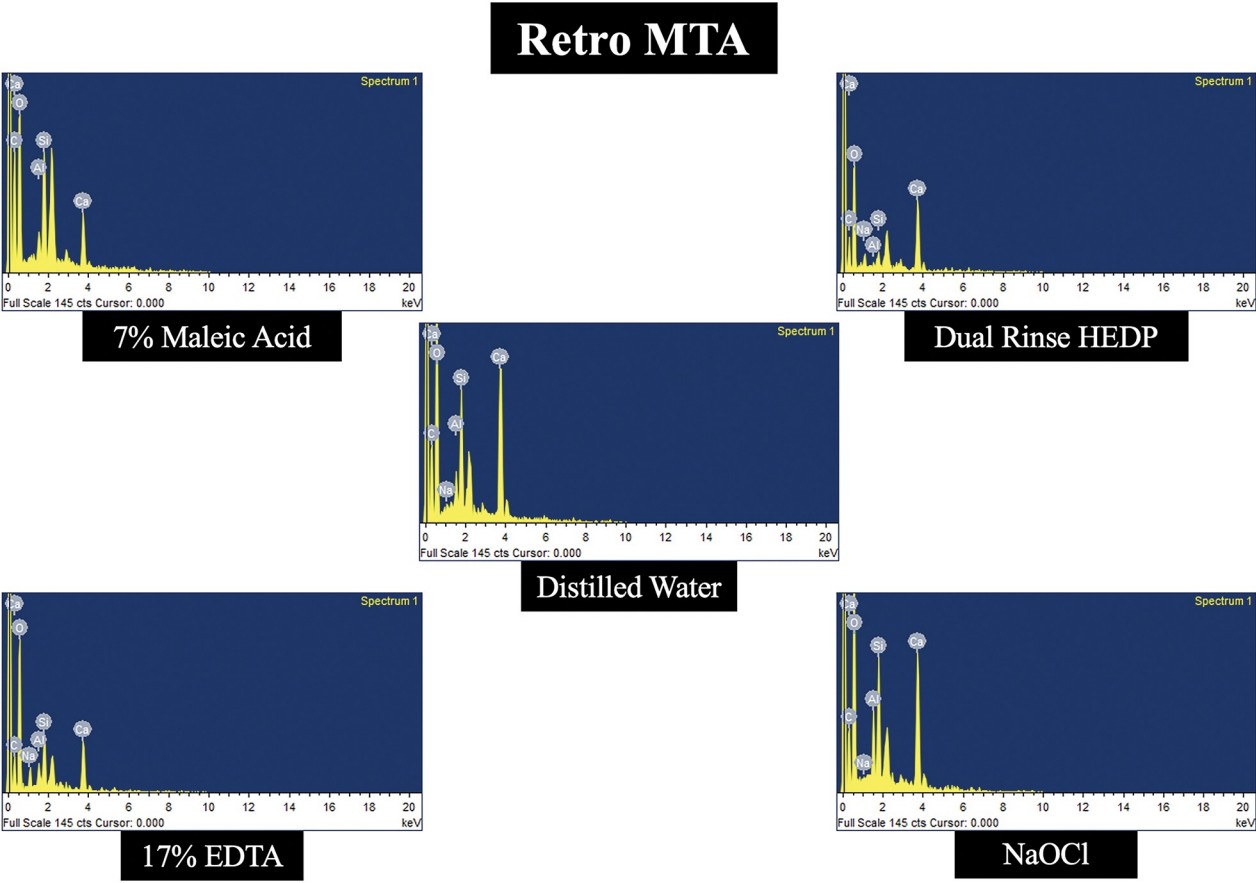

**Fig 5. Representative energy dispersive spectroscopy image of Retro-MTA cement treated with 7% MA, DR HEDP, 17% EDTA, 2.5% NaOCl and distilled water (control).**

HEDP did not alter the mineral components of the set MTA cements used in the present study.

Based on this result, when the perforation or resorption site is repaired using Ortho MTA or Pro-Root MTA, it is judicious to irrigate the root canal system using an etidronic acid-based chelating solution (DR HEDP) for the smear layer removal. However, the combination of HEDP with NaOCl had less detrimental effect on the adhesion of different MTA cements and their microstructure, the smear layer removal ability of this combination irrigating agent is inferior when compared to 17% EDTA [30]. In the current research, the samples filled with tricalcium silicate cements were wrapped in wet gauze for the complete set of the cement, which was similar to the method used by previous studies [31,32]. One of the limitations of the present study was the technique used to treat the MTA samples with experimental irrigating agents. The dentine discs filled with MTA cements were completely dipped in irrigating agents, which do not directly simulate the real clinical conditions. Clinically, only the coronal surface of the filling material is much more exposed to the irrigation solutions compared to the other surfaces, after perforation repair or apexification procedures [33].

## Conclusions

Within the limitations of this *ex-vivo* study, it can be concluded that when the tricalcium silicate cements are used in the perforation or resorption repair and apexification procedures, it

can be logical to use mild chelating agent like DR HEDP for smear layer removal, which do not have detrimental effect on bonding of tricalcium silicate cements to root dentine and its microstructure. This issue may prevent failures in the dislocation resistance of tricalcium silicate cements from the root canal dentine in the repaired area. Further studies conducted in real time clinical conditions are required, to assess the influence of different chelating agents on the longevity of tricalcium silicate cements used during endodontic treatment.

## Supporting information

**S1 Data.**
(XLS)

## Acknowledgments

The authors thank Medcem, GmbH, Weinfelden, Switzerland, for providing Dual Rinse HEDP for this study and Ajman University, Ajman, (UAE) for their continuous support.

## Author Contributions

**Conceptualization:** Özgür İlke Ulusoy, Nidambur Vasudev Ballal, Krishna Prasad Shetty, Alexander Maniangat Luke.

**Data curation:** Nidambur Vasudev Ballal, Nuran Ulusoy, Krishna Prasad Shetty, Alexander Maniangat Luke.

**Formal analysis:** Rajkumar Narkedamalli.

**Investigation:** Nidambur Vasudev Ballal.

**Methodology:** Nidambur Vasudev Ballal, Rajkumar Narkedamalli, Alexander Maniangat Luke.

**Project administration:** Özgür İlke Ulusoy, Nidambur Vasudev Ballal, Krishna Prasad Shetty.

**Supervision:** Özgür İlke Ulusoy, Nidambur Vasudev Ballal.

**Validation:** Nidambur Vasudev Ballal, Nuran Ulusoy.

**Writing – original draft:** Özgür İlke Ulusoy, Nidambur Vasudev Ballal, Nuran Ulusoy, Krishna Prasad Shetty, Alexander Maniangat Luke.

**Writing – review & editing:** Özgür İlke Ulusoy, Nidambur Vasudev Ballal, Krishna Prasad Shetty.

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
