## [Decision Letter · Decision Letter 0]

21 Aug 2023

PONE-D-23-13268Dislodgement resistance and Structural changes of tricalcium silicate-based cements after exposure to different chelating agentsPLOS ONE

Dear Dr.Shetty ,

Thank you for submitting your manuscript to PLOS ONE. After careful consideration, we feel that it has merit but does not fully meet PLOS ONE’s publication criteria as it currently stands. Therefore, we invite you to submit a revised version of the manuscript that addresses the points raised during the review process.

We look forward to receiving your revised manuscript.

Kind regards,

Fahad Umer

Academic Editor

PLOS ONE

Journal Requirements:

"NO"

"NO authors have competing interests"

4. We note that you have stated that you will provide repository information for your data at acceptance. Should your manuscript be accepted for publication, we will hold it until you provide the relevant accession numbers or DOIs necessary to access your data. If you wish to make changes to your Data Availability statement, please describe these changes in your cover letter and we will update your Data Availability statement to reflect the information you provide

Reviewers' comments:

Reviewer's Responses to Questions

**Comments to the Author**

1. Is the manuscript technically sound, and do the data support the conclusions?

Reviewer #1: Yes

Reviewer #2: Partly

2. Has the statistical analysis been performed appropriately and rigorously? 

Reviewer #1: Yes

Reviewer #2: Yes

3. Have the authors made all data underlying the findings in their manuscript fully available?

Reviewer #1: Yes

Reviewer #2: Yes

4. Is the manuscript presented in an intelligible fashion and written in standard English?

Reviewer #1: Yes

Reviewer #2: Yes

5. Review Comments to the Author

Reviewer #1: Thank you very much for a very nice manuscript. I congratulate the authors on their hard work. I have very few comments as follows

Introduction:

Line 68 ProRoot MTA and OrthoMTA are old products. They you have written it, it seems they are new. Please rephrase.

Line 94: Correct HEBP to HEDP

Results:

Table legends are not adequate. Please indicate what does the two and three Asterix imply. Also what test was used should be indicated.

Discussion:

The Combination product NaOCl/HEDP may not be as effective as EDTA at smear removal. You need to explore this more in your discussion, since any benefit that one may get from HEDP in terms of reduced detrimental effects on MTA are negated by the lack of or poor primary function i.e. the smear removal.

Reviewer #2: Except that the study was underpowered, all other technical aspects were taken care of and the statistical analysis was adequately performed. The introduction section is quite generic but the rest of the manuscript is adequately written otherwise and the methodology is quite crisp and easy to replicate.

6. PLOS authors have the option to publish the peer review history of their article (what does this mean?). If published, this will include your full peer review and any attached files.

Reviewer #1: **Yes: **Arshad Hasan

Reviewer #2: **Yes: **Nighat Naved

---

## [Author Response · Author response to Decision Letter 0]

22 Nov 2023

Manuscript has been revised and submitted

---

## [Decision Letter · Decision Letter 1]

18 Dec 2023

Dislodgement resistance and Structural changes of tricalcium silicate-based cements after exposure to different chelating agents

PONE-D-23-13268R1

Dear Dr.Shetty

We’re pleased to inform you that your manuscript has been judged scientifically suitable for publication and will be formally accepted for publication once it meets all outstanding technical requirements.

Kind regards,

Fahad Umer

Academic Editor

PLOS ONE

Additional Editor Comments (optional):

Reviewers' comments:

Reviewer's Responses to Questions

**Comments to the Author**

1. If the authors have adequately addressed your comments raised in a previous round of review and you feel that this manuscript is now acceptable for publication, you may indicate that here to bypass the “Comments to the Author” section, enter your conflict of interest statement in the “Confidential to Editor” section, and submit your "Accept" recommendation.

Reviewer #1: All comments have been addressed

2. Is the manuscript technically sound, and do the data support the conclusions?

Reviewer #1: Yes

3. Has the statistical analysis been performed appropriately and rigorously? 

Reviewer #1: Yes

4. Have the authors made all data underlying the findings in their manuscript fully available?

Reviewer #1: Yes

5. Is the manuscript presented in an intelligible fashion and written in standard English?

Reviewer #1: Yes

6. Review Comments to the Author

Reviewer #1: Thank you very much for a much revised paper. The manuscript in its revised form seems acceptable. Thank you

7. PLOS authors have the option to publish the peer review history of their article (what does this mean?). If published, this will include your full peer review and any attached files.

Reviewer #1: **Yes: **Arshad Hasan

---

## [Editor Report · Acceptance letter]

8 Jan 2024

PONE-D-23-13268R1 

PLOS ONE

Dear Dr. Shetty, 

I'm pleased to inform you that your manuscript has been deemed suitable for publication in PLOS ONE. Congratulations! Your manuscript is now being handed over to our production team.

Kind regards, 

on behalf of

Dr. Fahad Umer 

Academic Editor

PLOS ONE